# Biomedical Ti-Nb-Zr Foams Prepared by Means of Thermal Dealloying Process and Electrochemical Modification

**DOI:** 10.3390/ma15062130

**Published:** 2022-03-14

**Authors:** Grzegorz Adamek, Adam Junka, Przemyslaw Wirstlein, Mieczyslawa U. Jurczyk, Piotr Siwak, Jeremiasz Koper, Jaroslaw Jakubowicz

**Affiliations:** 1Institute of Materials Science and Engineering, Poznan University of Technology, Jana Pawla II 24, 61-138 Poznan, Poland; jaroslaw.jakubowicz@put.poznan.pl; 2Department of Pharmaceutical Microbiology and Parasitology, Wroclaw Medical University, Borowska 211, 50-534 Wroclaw, Poland; adam.junka@umed.wroc.pl; 3Department of Gynecology and Obstetrics, Division of Reproduction, Poznan University of Medical Sciences, Polna 33, 60-535 Poznan, Poland; abys@wp.pl; 4Division Mother’s and Child’s Health, Poznan University of Medical Sciences, Polna 33, 60-535 Poznan, Poland; mjur@poczta.onet.pl; 5Institute of Mechanical Technology, Poznan University of Technology, Piotrowo 3, 60-965 Poznan, Poland; piotr.siwak@put.poznan.pl; 6Electrotile Sp. z o.o., Puławska 427, 02-801 Warszawa, Poland; koperjeremiasz@gmail.com

**Keywords:** titanium-based metallic foams, mechanical alloying, thermal dealloying

## Abstract

The paper presents results of preparation and modification of Ti20Nb5Zr foams by a thermal dealloying method followed by electrochemical modification. The first step of this study was the preparation of Ti20Nb5Zr30Mg nanopowder using mechanical alloying (MA). The second was forming green compacts by cold pressing and then sintering with magnesium dealloyed from the structure, which resulted in pores formation. The next step was surface modification by electrochemical etching and silver nanoparticle deposition. Porosity, morphology, mechanical properties as well as biocompatibility and antibacterial behavior were investigated. Titanium foam porosity up to approximately 60% and wide pore size distribution were successfully prepared. The new materials have shown positive behavior in the MTT assay as well as antibacterial properties. These results confirmed great potential for thermal dealloying in preparation of porous structures.

## 1. Introduction

Titanium alloys are commonly used in hard tissue implant applications due to their excellent corrosion resistance, relatively good mechanical properties and superior biocompatibility. However, metallic biomaterials including titanium alloys have much higher elastic moduli than bones, which may result in implant loosening [1,2]. There are two ways to reduce the stiffness: by the right combination of chemical composition and, more effectively, by preparation of porosity in the material which could then be named a metallic foam. Such a form of material could act like a scaffold for the tissue and can exhibit a unique combination of properties, especially mechanical. Moreover, open spaces lead to bone tissue in-growth and, when they are interconnected, they enhance body fluid transportation [3,4,5,6,7,8]. Nowadays, alloying elements for titanium should be biocompatible, corrosion resistant and should stabilize the beta structure, which shows better Young’s moduli. Such elements are Zr, Nb and Ta. They belong to the nontoxic and non-allergic group of elements [9,10,11,12,13]. Niobium and tantalum are considered to be the strongest beta stabilizers. Zirconium, if taken separately, is considered a neutral or weak beta stabilization element. However, Zr begins to play a role of an effective beta stabilizer in the presence of Nb or Ta [14,15,16,17]. Ti-Nb-Zr alloys are still extensively researched and are the most attractive alloys for biomedical application due to their biological and mechanical properties, with Young’s moduli and yield strength ranging from 50 to 85 GPa and from 350 to over 900 MPa, respectively. Some of the alloys show a shape memory effect and superelasticity at room temperature [18,19,20,21,22,23,24,25,26].

In their previous papers, the authors described a procedure of a new metallic foam preparation method by removing Mg from the alloy. It was called thermal dealloying of magnesium [27,28]. In this method, we proposed to prepare a titanium-based alloy with the right amount of magnesium by mechanical alloying (MA). In the sintering stage, when the temperature rise above the boiling temperature of Mg, the Mg atoms separated from the alloy, diffused from the middle of the sample to the surface and evaporated, leaving pores. Mg has a relatively low boiling temperature (1091 °C) and is also biocompatible—so even if some Mg remains in the alloy, that is not a problem.

In this paper, we would like to go further with foam preparation and apply electrochemical modification to them. Surface modification of biomaterials is often used for improvement of biocompatibility. Different methods were applied by other authors to modify metallic foams [29], i.e., electrophoretic deposition of calcium phosphate/chitosan layer [30], PVD metallic coatings [31], sol-gel dip coating of polymer [32], PLGA layer deposition by vacuum infiltration [33], micro-arc oxidation and calcium phosphate coatings preparation [34] and electrochemical deposition of hydroxyapatite [35]. Moreover, Vadim et al. modified Ti-Nb-Zr foams by immersion in a polymer solution [36], and Lascano et al. used CVD to transfer graphene on the Ti-Nb-Ta-Mn porous alloys [37]. However, we could not find in the literature an investigation on electrochemical etching of titanium-based foams. Electrochemical etching for biomedical alloys results in surface pores and oxide formation, which improves bone fixing and corrosion resistance and could be even more effective for sintered materials [38]. Another example of surface modification could be electrochemical deposition of Ag nanoparticles. The use of silver nanoparticles, as an implant’s surface modifier, is often tested to improve antibacterial properties and decrease the risk of postoperative infection [39,40,41]. Modification methods have influence on biological properties and could improve osseointegration [29].

In this work, we decided to investigate Ti20Nb5Zr foams prepared by mechanical alloying (MA) with addition of 30% of Mg, followed by thermal dealloying. The Ti-Nb-Zr alloys are well-described in the literature and show good properties from a biomaterials point of view [18,19,20,21,22,23]. One of the goals of this study was to investigate the preparation of such alloys by MA in the presence of Mg. Moreover, the aim of this work was to develop metallic foam by means of thermal dealloying and electrochemical modification.

## 2. Materials and Methods

### 2.1. Foam Preparation

The Ti20Nb5Zr30Mg alloy was prepared by MA under an argon atmosphere using a SPEX 8000 Mixer (SPEX SamplePrep., Metuchen, NJ, USA) Mill with the ball-to-powder weight ratio of 20:1 (ball weight 50 g, 5 item, powder weight 2.5 g). The initial powders were Ti (purity 99.5%), Nb (purity 99.98%), Mg (purity 99.8%), all 325 mesh from AlfaAesar and Zr shavings from the rod (purity ≥99%) from Aldrich. The weight of the starting powder was measured by precision balance (0.001 g repeatability; Radwag). Based on our previous studies, we decided to investigate the alloy with 30% of Mg, which corresponded to 60% of final foam porosity [27]. All operations on powders were performed in a glove box (LabMaster 130, MBRAUN, Garching, Germany) filled with an automatically controlled argon atmosphere (O_2_ < 2 ppm and H_2_O < 1 ppm). The milling process was stopped after a certain period of time to investigate structure and powder morphology changes, which was performed by XRD (Panalytical, Malvern, UK) and SEM (Tescan, Brno, Czech Republic). The milling time was designed to achieve a fine beta Ti structure and was 144 ks for the investigated composition. The consolidation stage consisted of two parts. First, the powder was portioned by placing into a steel die (diameter 8 mm) and uniaxially pressing at a pressure of 1000 MPa. The green compacts were 4–5 mm in height. The second step was sintering/dealloying, which was carried out in a tube furnace (Nabertherm, Lilentahl, Germany) at 1300 °C for 14.4 ks. To remove the magnesium vapors from the material and prevent excessive oxidation, the sintering/dealloying was carried out in a 10^−2^ Pa vacuum. Then, the sinters were slowly chilled to room temperature together with the furnace.

### 2.2. Electrochemical Modification of Foams

The prepared foams were than modified by two electrochemical treatments: electrochemical etching and electrochemical silver deposition. The process was controlled by a Solartron 1285 potentiostat (Solartron Analytical, Farnborough, UK). A Gamry Multiport Corrosion Cell Kit with SCE as the reference and graphite as the counter electrodes was used. The electrolyte was mixed using magnetic stirring (250 rpm). The electrochemical etching parameters were as follows: electrolyte 1 M H_3_PO_4_ + 0.1 M HF (POCH S.A., Poland), potential 10 V vs. OCP (open circuit potential), time 300 s, temperature 22 °C. The electrochemical Ag deposition parameters were as follows: electrolyte 0.01 M AgNO_3_ + 0.01 M HNO_3_ (POCH S.A. Poland), potential −1 V vs. OCP, time 15 and 60 s, temperature 22 °C. Additionally, an Ag plate (5 cm^2^) was used to support the transport of the Ag ions and Ag refilling into the electrolyte. After the processes, the samples were rinsed in distilled water and dried in a stream of nitrogen.

### 2.3. Structural and Microstructural Analysis

The structure and phase analyses were investigated using Empyrean XRD (Panalytical, Malvern, UK) with CuKα radiation and High-Score Plus software. Microstructure was determined by Mira 3 FEG SEM (Tescan, Brno, Czech Republic) equipped with In-Beam SE and an EDS UltimMax 65 (Oxford Instruments, Abingdon, UK) detector. The porosity level was examined by a GX51 (Olympus, Tokyo, Japan) optical microscope on polished, unetched samples and image analysis by Stream Start and GIMP software. On such images, the white and black areas corresponded to polished material and pores, respectively. The amount of black color (percentage from color histogram) was in accordance with the porosity level.

### 2.4. Mechanical Measurements

The nanohardness measurements were made with a Picodentor HM500 (Helmut Fisher, Hartford, CT, USA) nanoindenter. According to ISO 14577-1 standard [42] the HV-Vickers Hardness and EIT-indentation modulus were measured. The indentations were made at the force of 300 mN for 20 s. A 4483 Instron mechanical testing machine (Instron, Norwood, MA, USA) with a strain rate of 0.001 s^−1^ was used to measure the compressive strength and elastic moduli.

### 2.5. MTTAssay

Normal human osteoblasts (CC-2538) (NHost) and human periodontal ligament fibroblasts (CC-7049) (HPLF) (Lonza Group Ltd., Basel, Switzerland) were used for the in vitro cytocompatibility assay. The tests’ parameters were as follows: conditions static; cultured cells concentration 5000 cells/well in 1 mL of culture medium; temperature 37 °C; atmosphere 5% CO_2_; incubating time 24, 48, 72 and 96 h. The proliferation of the cells in the conditioned mediums was expressed as a percentage of the value of relative viability of the cells (RVC) of the reference medium. The reference media were prepared using pure bulk microcrystalline titanium samples and was represented by 100%. All samples were sterilized (autoclaving for 15 min at 120 °C) and separately located in 24-well microplates. Kruskal–Wallis one-way Analysis of Variance on Ranks with multiple repetition option SigmaStat 3.5 (Systat Software Inc., Chicago, IL, USA) with U Mann–Whitney test was used for statistical significance analysis. The significance level was *p*-value < 0.05.

### 2.6. Antibacterial Properties

The following ATCC collection strains were chosen for experimental purposes: *S. aureus* 6538, *P. aeruginosa* 15,442 and *C. albicans* 10,231. In biofilm formation experiments, all strains were cultured on TSB liquid media (Biocorp, Warsaw Poland), and appropriate agar media (Columbia, Sabouraud, McConkey, Biocorp, Warsaw Poland) were used in strain preservation. The foams and cp-Ti in the form of discs were the study and control groups, respectively. The strains cultured on appropriate agar plates (*S. aureus*, Columbia plate; *C. albicans*, Sabourad plate; *P. aeruginosa*, McConkey plate) were transferred to liquid TSB medium and incubated for 24 h at 37 °C under aerobic conditions. After incubation, the strains were diluted to the density of 1 McFarland (MF) using a densitometer (Densi-LA-meter II, Biosciences, Riverside, CA, USA). The microbial dilutions were inoculated to the wells of 24-well plates (VWR, Gdansk, Poland) containing Ag or C titanium alloy discs. Another 24 h/37 °C incubation was performed. Next, the discs were rinsed using sterile saline to remove unbound cells and to leave the biofilm on the disc surface only. Afterwards, the discs with biofilm were transferred to fresh, sterile TSB medium containing 1% 2-, 3-, 5-triphenyl-2H-tetrazolium chloride (TTC. Sigma-Aldrich, Darmstadt, Germany) and left for 4 h. TTC is a colorless compound that changes into red formazan in the presence of living, metabolically active microbial cells. After incubation, the discs were transferred to an ethanol/acetic acid mixture 95:5 (vol/vol) and shaken vigorously on the plate-shaker (Schuttiken, Germany) to release dye from the cells. Subsequently, suspension in medium formazan was collected, and its concentration was measured using a spectrometer (Thermo Scientific Multiscan, GO, USA) at a wavelength of 490 nm. Statistical analysis was performed using statistical calculations performed with the SigmaStat package, Version 2.0 (SPSS, Chicago, IL, USA). Quantitative data from the experimental results were analyzed using an unpaired *t*-test with Welch’s correction with the significance level established at *p* < 0.05.

## 3. Results and Discussion

In this paper, the authors present the procedure of Ti-based foam preparation carried out in two steps: mechanical alloying followed by a sintering/dealloying process. In Figure 1, there are data showing changes in the phase constitution during MA (observed using XRD) and the morphology of the powders (SEM images). In this study, a Ti20Nb5Zr30Mg alloy was investigated. The main goal of the first step of foam preparation was to dissolve Mg in the Ti structure. Mechanical alloying gives great opportunity to achieve it. Moreover, other elements were dissolved in the Ti structure during milling, resulting in phase transformation to Ti-β. After 1 h of milling, there were well-visible peaks of alloying elements Ti, Mg, Nb and Zr. In the first stage of MA, a mixture of the elements was formed. Decreases in Zr, Nb and Mg peaks’ intensity was observed over time: after 10 h of milling, there was no visible peak of Zr, and after 30 h, the last peak of Mg (2 theta 35th degree) was almost totally reduced. Peaks form Nb (beta-stabilized element) were close to the Ti-β and on diffractograms and it appeared that the intensity decreased really slowly. However, the phase transformation from Ti-α to Ti-β took place here with additional fragmentation of the microstructure. It resulted in peaks corresponding to only Ti-β after 40 h of milling for the investigated alloy. Our previous work proved that the time needed to achieve a one-phase structure depended on chemical composition and in some cases could be ineffectively long [27,28]. At different stages of MA, the powders could agglomerate or fracture. For the Ti20Nb5Zr30Mg alloy, we could observe a classic occurrence: after a short time of milling, the powders agglomerated to quite large dimensions of about 200 µm, and with increasing time, the size of the particles decreased linearly to about 25–50 µm. However, the average crystal size after 40 h was 51 ± 8 nm. The obtained nanostructure was helpful for the dealloying stage, because of the large number of grain boundaries, which was the easiest way leading to the diffusion and evaporation of Mg in high-temperature consolidation. The morphology of the powders at the subsequent MA stages shown in Figure 1 was uniformly fractured. Here, as well as for the other mechanically alloyed titanium and tantalum alloys containing magnesium [27,28] investigated by the authors, great yield characteristics were observed. After 40 h of the MA process, more than 92% of the powder yield was achieved.

Figure 2 shows the scheme made up of optical micrographs of the microstructural mechanism of pore formation at various temperatures. To prepare this scheme, we stopped each sintering process at subsequent temperatures and chilled down right away (500, 800, 1000 and 1100 °C). The sintering of the sample at 1300 °C took 4 h. After cold pressing, even if pressure as high as 1000 MPa was used, there were some pores in the microstructure. In temperatures below the boiling point of Mg (Figure 2, 500 and 800 °C) the typical mechanisms of densification took place, resulting in decrease of porosity from 4.8 to 1.4%, respectively, with characteristic more rounded pores.

In this study, we found that the dealloying process started below the boiling temperature of Mg. In Figure 2 (1000 °C) there are clearly visible relatively large pores located on particle boundaries as well as smaller ones inside them. It is well-known that a nanostructure could decrease characteristic temperatures of materials and here we could probably observe the effect of Mg boiling temperature decrease. The porosity level reached 25.7%. It increased with increasing temperature (38.3%/1100 °C) and finally reached 58.6% after sintering at 1300 °C for 4 h. In such a high temperature, the Mg atoms started to separate from the alloy, and the dealloying process started. With time, the atoms diffused from the middle to the surface and combined to form vapors and then evaporate (this stage was performed in a vacuum), leaving open spaces—pits and pores, which were mostly interconnected and were characterized by a relatively wide range of size distribution (from nano- to micro-scale). SEM micrographs of the polished sample prepared by 40 h MA followed by dealloying at 1300 °C/4 h are shown in Figure 3. In Figure 3a, we can observe pores as well as flat polish areas. There were two types of pores visible at different magnifications. In Figure 3b–d, we can see numerous relatively large pores (from 1 to over 100 µm) interconnected, winding in shape. The second type of pores was much smaller (approximately from 0.25 to 1 µm) and regular in shape, located mostly in areas between the larger pores—inside the primary particles. They were also interconnected. The total porosity was approximately 58%, which nicely corresponded to the designed value (60%). Pore size distribution is shown in Figure 4. The trend line confirms the existence of two types of pores, and the relation of smaller to bigger was 48.3/51.7, so almost half and half. In comparison to those of Ti30Ta, Ta20Ti and Ta30Ti alloys prepared by thermal dealloying of Mg, the current results seemed to be similar in achieved pore size and shape. Moreover, the pore size distribution was similar as well, with range from nano- to micro-scale. However, for the Ti-30Ta foam, a bigger participation of smaller pores (from 0.02 to 1.00 µm) was observed [27,28]. Other authors prepared Ti-Nb-Zr foams using the space-holder technique. Aguilar et.al. [19] prepared Ti13Zr13Ta3Nb foams using NaCl as a space-holder. Their material had pore size distribution from 20 to 600 µm. Another often-used spacer is ammonium bicarbonate. Lascano et al. [37] investigated Ti35Nb-29Ta-xMn with a pore size from 20 to 800 µm. On the other hand, Rao et al. [23] prepared Ti20Nb15Zr and Ti35Nb15Zr foams using NH_4_HCO_3_ with pores corresponding in size and shape to spacer particles with a lower range limit of 50 µm. Brailovski et al. [10] showed properties of Ti20.5Nb5.6Zr scaffolds obtained by means of evaporation of a polymeric foaming agent with pore size distribution from 30 to over 1900 µm. When comparing thermal dealloying with the space-holder technique, we could observe a possibility to obtain smaller, including nano-scale, pores by the method proposed in this study. Moreover, the larger pore walls were relatively more developed in this case.

Our previous experience with foams prepared by thermal dealloying of Mg showed negligibility of the impact of chemical composition on mechanical properties like compression strength or elastic moduli in highly porous materials [27,28]. This conclusion was confirmed in the present paper. The properties of the Ti20Nb5Zr foam (approximately 60% porosity) were as follows: compression strength 15.5 ± 0.9 MPa, elastic moduli 0.71 ± 0.1 GPa, 98 ± 5 GPa measured by compression tests and using a nanoindentation tester, respectively, HV hardness 276 ± 20, calculated yield strength 92 MPa (σ = HV/3) [43,44]. Figure 5 shows the stress-strain curve recorded for the Ti20Nb5Zr foam. The shape of the stress-strain curve showed two characteristic regions. At initial raising strain, the elastic deformation region was observed. The second region was the stress plateau. However, the plateau was not flat and did not have a constant value. Stress oscillation in this part of the curve was due to pore collapsing and scaffold cracking during sample compression.

The main factor influencing the properties measured by compression tests is the porosity level, pore size, shape and distribution. The nanoindentation test is more local, the influence of porosity is limited, so we could find more information about the alloy itself. It is well-known that hardness, moduli and yield strength are structure-sensitive properties, and they can by influenced by, inter alia, chemical composition and grain size. Due to solid solution strengthening as well as fine microstructure, the hardness was higher than that of pure titanium. However, the calculated yield strength was three times lower than that of pure Ti and four times lower than that of bulk annealed Ti20.5Nb5.6Zr [10]. In the biomaterials-related aspect, the properties of the whole sample seemed to be more important, especially when the porosity determines mechanical properties. The moduli were close to that of human cancellous bone (from 0.1 to 2 GPa), so such materials have promising application potential [8].

In this study, we wanted to go further with foam preparation and decided to apply some electrochemical modification in two different ways: electrochemical etching in electrolyte consisting of HF as well as electrochemical deposition of Ag nanoparticles. Both have influence on the biological properties and could improve them. Application of electrochemical etching for biomedical alloys results in the surface pores and oxide formation, which improves bone fixing and corrosion resistance. The large volume of grain boundaries as well as primary pores improves the etching process by facilitating the sinter electrolyte penetration, resulting in effective material removing and new pore forming [38]. The morphology of the etched sample is shown in Figure 6. The treatment proposed in this study affected the surface of larger pores with significant changes of the pore walls, due to short application time (300 s). The smaller pore morphology was dramatically changed in terms of pore size and shape (compare Figure 3f and Figure 6d). The surface was more developed compared to that of foam before etching and definitely more developed than those of other authors’ results [10,19,23,37]. The pore size was in the rage of 0.5–5 µm and definitely of round shape. Moreover, the new surface in some areas was quite smooth, and nanolamellar morphology could also be observed (Figure 6d). All of these could be helpful in biocompatibility improvement due to increasing surface development and interconnected porosity. Porosity leads to bone tissue in-growth, and interconnections enhance body fluid transportation [3,4,5,6].

On the other hand, antibacterial properties are also welcome when it comes to reducing the risk of postoperative bacterial infection. The deposition of Ag is an effective way to achieve that. In our previous work [45], we showed the possibility of Ag nanotrees deposition on porous scaffolds. In this paper, we wanted to go a step further in such treatment. Figure 7 shows the micrographs of the Ti20Nb5Zr foam with deposited silver particles. The most important parameter of the process is time. It influences the size and shape of the deposited particles. In Figure 7a, we can see the general view of the sample after deposition of 60 s. Such time results in both regular nanoparticles of 30–100 nm in size as well as larger, tree-like, elongated particles of 1–5 µm in size. We found that with increasing time, the Ag precipitates were growing. If we reduced the time to 15 s, the whole surface was covered only with particles like those visible in Figure 7d,e. Therefore, shorter deposition time resulted just in nucleation and thus in greater amount and surface area of silver precipitates.

The surface chemical composition after Ag deposition was investigated by EDS. The EDS analysis of the Ti20Nb5Zr foam with deposited silver particles, shown in Figure 8, confirmed the presence of silver on the sample surface as well as titanium and alloying elements. Moreover, the analysis did not indicate the presence of magnesium, which suggested completeness of the dealloying process.

Cytotoxic activity was analyzed using an MTT assay. The obtained RVC results for human periodontal ligament fibroblasts and normal human osteoblasts are shown in Figure 9a,b, respectively. The results revealed a significant effect of time as well as material condition on cell viability. After 24 h, the RVC for all the tested materials was lower (oscillating in the range of 50–70%) than that of the reference sample, which is often caused by longer adaptation time of cells to a richer chemical composition of new-material medium. For raw and electrochemically etched foams, the cytotoxicity clearly decreased over time. Such results allowed a conclusion that such foams were nontoxic. However, there were significant differences between raw and etched ones in all evaluation points. At 96 h, all materials showed the highest viability. The modified and unetched foams showed almost 1.56 and 1.28 times better RVC, respectively (HPLF).

The results for osteoblasts showed the same trend but lower RVC. It was 1.5 and 1.04 times that of the reference condition for etched and unetched materials, respectively. Foams with silver particles showed definitely lower RVC for both HPLF and NHost. The time of Ag deposition had a great influence on size, shape and number of deposited particles as well as cytotoxicity. Foam with nano-Ag (15 s) showed increasing RVC over time. However, the trend was significantly lower than for foams without Ag and did not reach the control sample value. The foam with a higher amount of Ag particles showed stable RVC only at 0.6 times that of the reference condition. It confirmed the fact that there was a fine line between Ag biocompatibility and cytotoxicity.

Antibacterial behavior of biomaterials is also important and developed nowadays. As can be observed in Figure 10, the biofilm of all tested microbial species formed less eagerly on the surface of Ag-modified discs in comparison to that of non-Ag-modified samples. In the case of biofilm formed by *S. aureus, P. aeruginosa* and *C. albicans*, these differences were statistically significant (*t*-test, *p* < 0.05). The raw as well as electrochemically etched foams showed similar behavior to bulk cp-titanium samples. As shown in Figure 10, the reduction of biofilm formation on Ag-modified discs was substantial but not complete. Therefore, we decided to examine surviving biofilm structures on Ag-modified foams vs. biofilm structures on control samples by means of SEM (Figure 11). SEM results confirmed the results of semi-quantitative analysis presented in Figure 10—biofilm formed on Ag-modified discs displayed lower density, and surfaces completely devoid of bacteria were visible (Figure 11a), whereas on the control samples, thick, dense biofilm structure covered virtually the whole disc surface (Figure 11b).

## 4. Conclusions

In this paper, a new method of foam preparation with electrochemically modified surface was shown. Such materials could find application in production of hard tissue implants. Based on the presented results, the following conclusions could be drawn:Using mechanical alloying and alloying elements like Mg, Nb and Zr in Ti20Nb5Zr30Mg stoichiometry, it was possible to achieve a beta titanium structure after 40 h of MA.It was confirmed that when using magnesium as the alloying element, cold welding did not have significant relevance for titanium alloys. More than 92% of powder yield was achieved.Mechanical alloying was beneficial in the synthesis of nanocrystalline Ti-Nb-Zr-Mg alloys.Thermal dealloying of magnesium resulted in the formation of a porous interconnected scaffold.Electrochemical treatment of prepared foams resulted in improvement of surface development and could improve biological and antibacterial properties.

## Figures and Tables

**Figure 1 materials-15-02130-f001:**
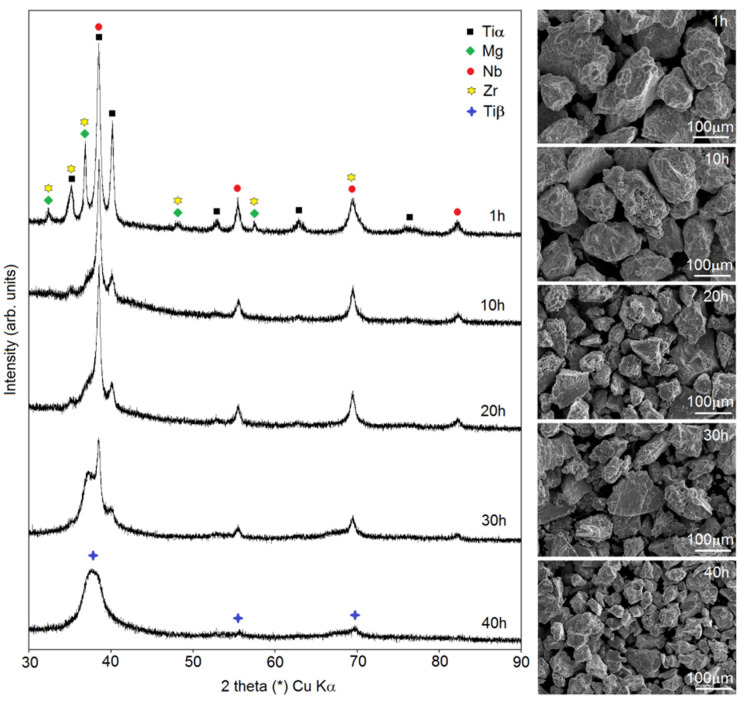
XRD spectra of the Ti20Nb5Zr30Mg and SEM micrographs of the powders at various milling times.

**Figure 2 materials-15-02130-f002:**
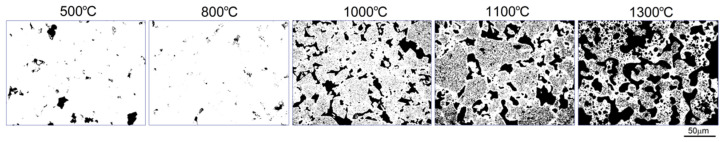
Scheme of Ti20Nb5Zr-Mg thermal dealloying process—optical micrographs at various stages.

**Figure 3 materials-15-02130-f003:**
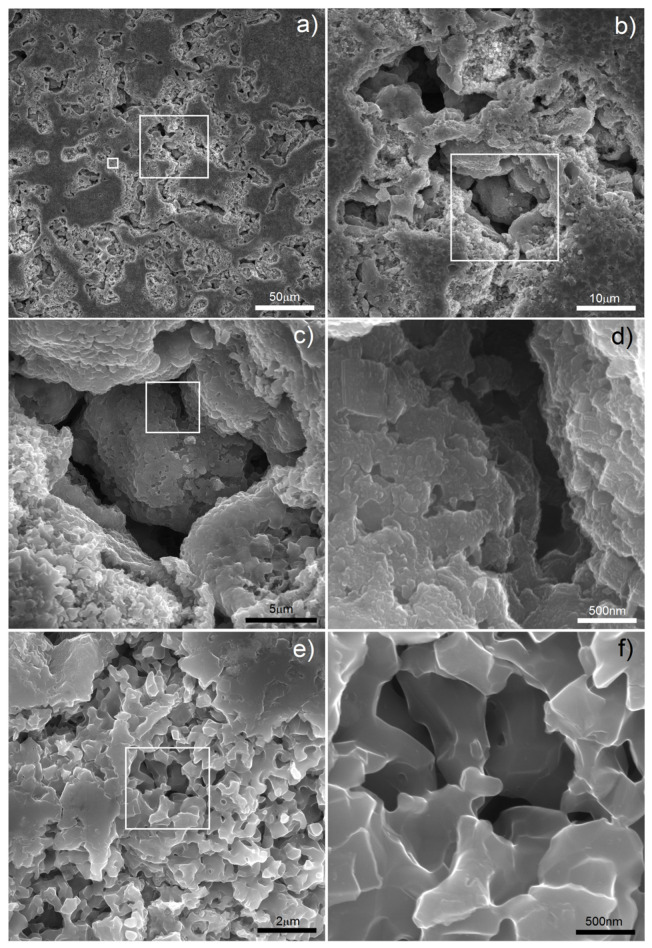
SEM micrographs of Ti20Nb5Zr foam after dealloying: (**a**–**d**) various magnifications of larger pores; (**e**,**f**) various magnifications of smaller pores.

**Figure 4 materials-15-02130-f004:**
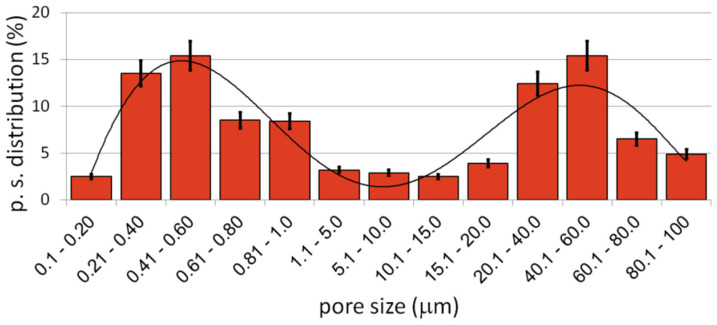
Ti20Nb5Zr foam pore size distribution.

**Figure 5 materials-15-02130-f005:**
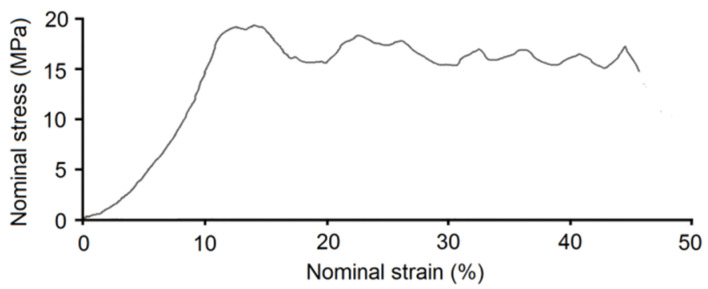
Stress-strain diagram of Ti20Nb5Zr foam.

**Figure 6 materials-15-02130-f006:**
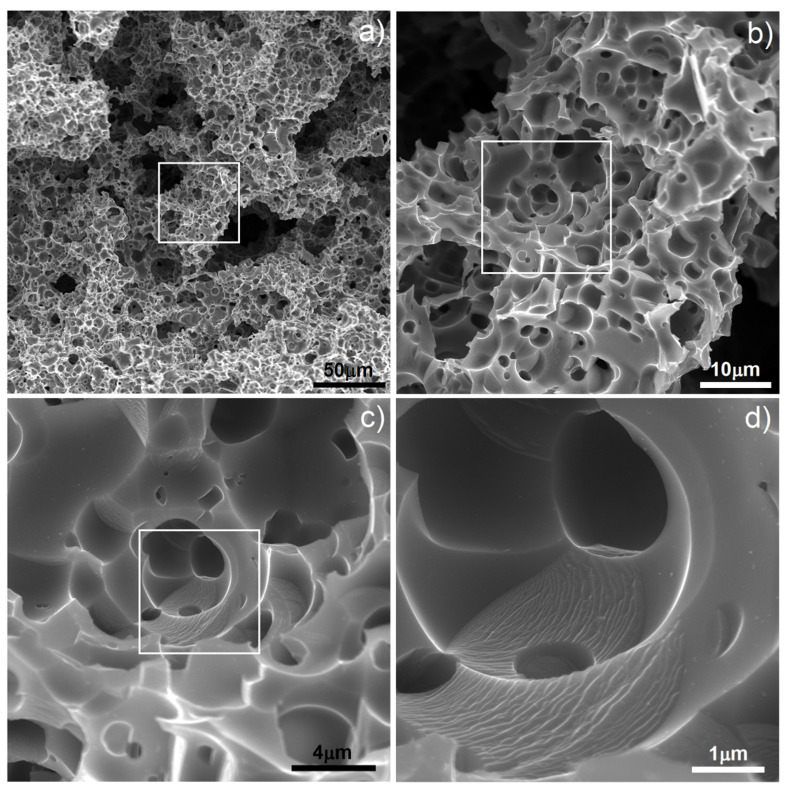
SEM micrographs of the Ti20Nb5Zr foam after electrochemical etching: (**a**–**d**) various magnifications.

**Figure 7 materials-15-02130-f007:**
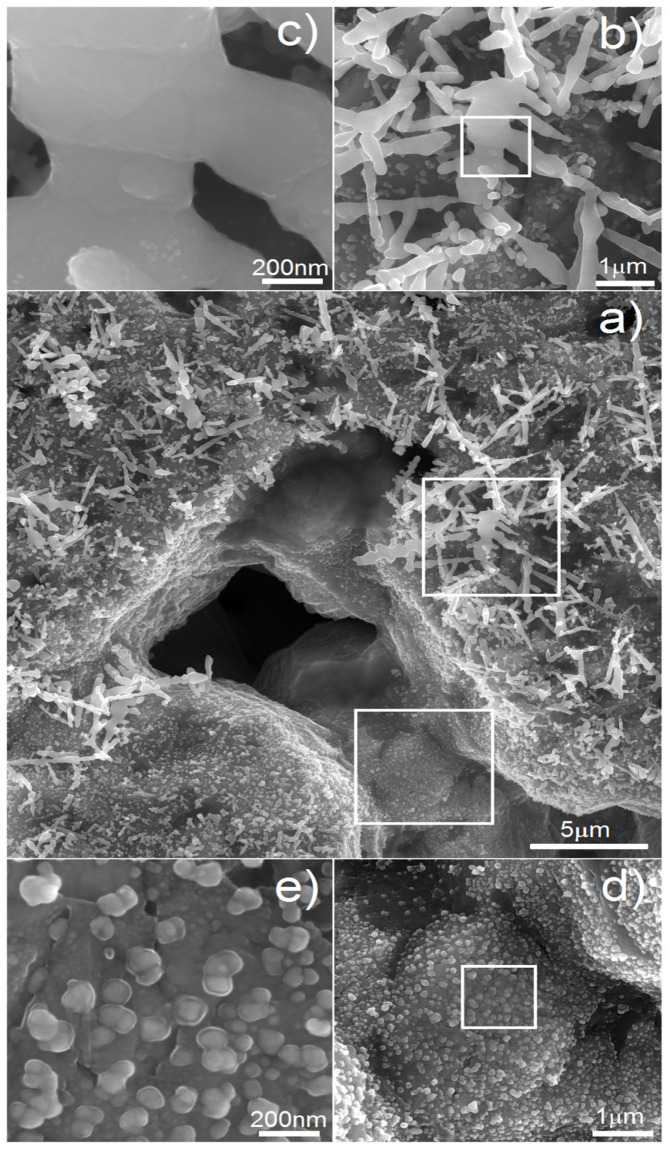
SEM micrographs of the Ti20Nb5Zr foam with electrochemically deposited Ag particles: (**a**) general view; (**b**,**c**) various magnifications of tree-like Ag structure; (**d**,**e**) various magnifications of Ag nanoparticles.

**Figure 8 materials-15-02130-f008:**
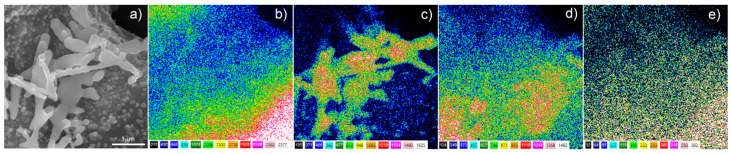
EDS mapping of the Ti20Nb5Zr +Ag foam: (**a**) SEM image; (**b**) Ti; (**c**) Ag; (**d**) Nb; (**e**) Zr. Lighter-colored areas correspond to higher concentration of the element.

**Figure 9 materials-15-02130-f009:**
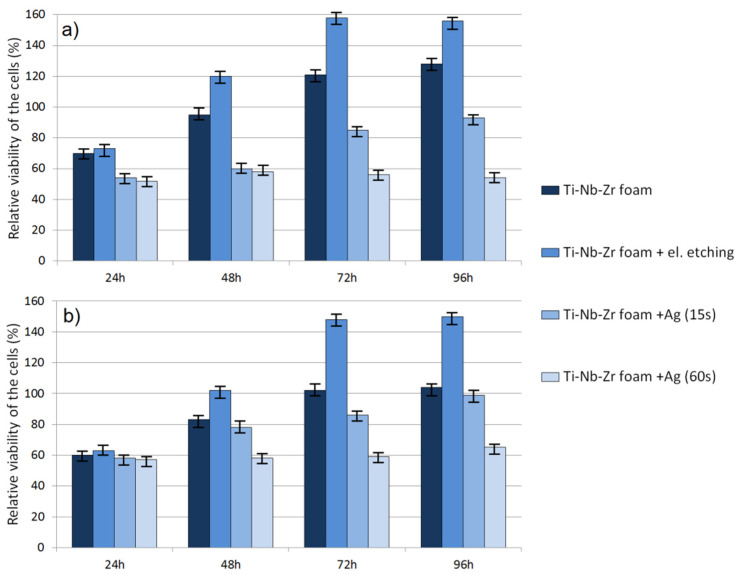
Results of the MTT assay performed at 24, 48, 72 and 96 h: (**a**) HPLF cells; (**b**) NHost cells.

**Figure 10 materials-15-02130-f010:**
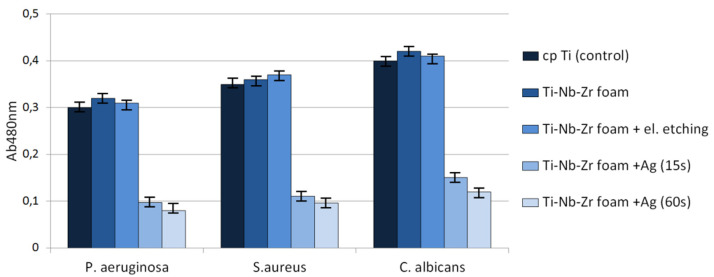
Reduced ability of tested microbes to form a biofilm on the foam surface.

**Figure 11 materials-15-02130-f011:**
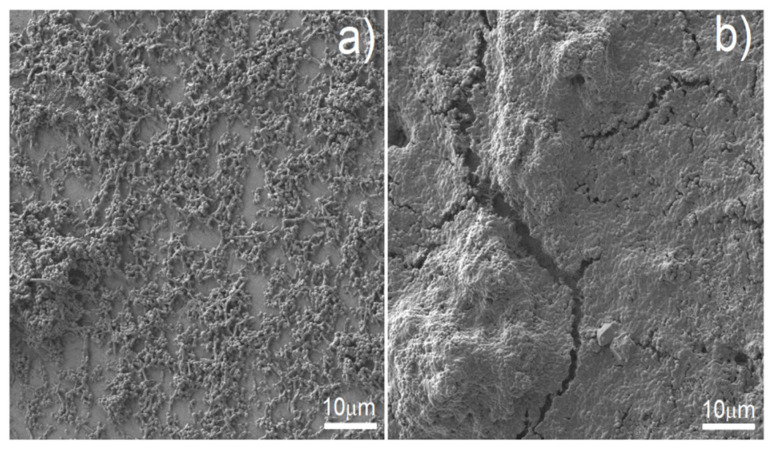
Pseudomonal biofilm formed on Ag-modified discs—particular cells are seen, they are not completely covered with biofilm matrix (**a**), and thick, multilayer pseudomonal biofilm formed on the control sample (**b**).

## Data Availability

Not applicable.

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
