# Peer review of "Biomedical Ti-Nb-Zr Foams Prepared by Means of Thermal Dealloying Process and Electrochemical Modification"

_materials, 2022, doi:10.3390/ma15062130_

Round 1
Reviewer 1 Report
The manuscripts tried to introduce the properties of the biomedical Ti-Nb-Zr foams prepared by means of the thermal dealloying process and electrochemical modification. The manuscript has considerable flaws, and I will reconsider the manuscript after a major revision. The following points should be considered in the revised version:
1- The introduction of the manuscript is not adequate to explain the essentiality of the study. Please add more relevant references and clarify the exact motivations for this study in a readable story.
2- In Fig. 7, the labels of elements are not clear. Please magnify that, and also, more explanation is needed to support these results. The connection to the rest of the results is not strong.
3- The English language of the manuscript is not appropriate for publication, and a revision is needed by an English native speaker expert.
4- The efficiency of the foam preparation by electrochemically modified surface should be discussed, and compared with the other methods with a quantifying method. To introduce a new method, a strong comparison with other methods is very important.
5- The mechanical properties of the foam were reported, and interesting results were obtained. However, the yield strength of the material is also very important, and by considering that, and the obtained elastic moduli, one can understand the extent of possible elastic strain to be able to impose before yielding. Based on the obtained results of hardness by indentation, you can calculate the yield strength, ?y=??/3. Please cite this reference, https://doi.org/10.1002/srin.202000242, and calculate and extend the discussion.
6- In lines 228-232, some conclusions were obtained about effective material removal and new pores formin, however, it is not clear how the authors came to these conclusions. please explain that with sufficient support by references and your results.
Author Response
We would like to express our great appreciation to the Editor and respected Reviewers for examining the manuscript carefully and providing us valuable comments and constructive suggestions which are very important for the revision of the manuscript and improving the quality of the work. We have tried our best to revise our manuscript according to their suggestions.
Replies to Reviewer #1:
Reviewer #1:
Comment 1: The introduction of the manuscript is not adequate to explain the essentiality of the study. Please add more relevant references and clarify the exact motivations for this study in a readable story.
Reply: The introduction was improved and we add more relevant references.
Comment 2: In Fig. 7, the labels of elements are not clear. Please magnify that, and also, more explanation is needed to support these results. The connection to the rest of the results is not strong.
Reply: The figure was changed and explanation was added in figure caption.
Comment 3: The English language of the manuscript is not appropriate for publication, and a revision is needed by an English native speaker expert.
Reply: The English language was revised by professional translator.
Comment 4: The efficiency of the foam preparation by electrochemically modified surface should be discussed, and compared with the other methods with a quantifying method. To introduce a new method, a strong comparison with other methods is very important.
Reply: improved
Comment 5: The mechanical properties of the foam were reported, and interesting results were obtained. However, the yield strength of the material is also very important, and by considering that, and the obtained elastic moduli, one can understand the extent of possible elastic strain to be able to impose before yielding. Based on the obtained results of hardness by indentation, you can calculate the yield strength, ?y=??/3. Please cite this reference, https://doi.org/10.1002/srin.202000242, and calculate and extend the discussion.
Reply: the yield strength was calculated according to Your suggestion and discuss, we cite Your publication, however in our opinion it does not fit well to the manuscript, so we add another one.
Comment 6: In lines 228-232, some conclusions were obtained about effective material removal and new pores formin, however, it is not clear how the authors came to these conclusions. please explain that with sufficient support by references and your results.
Reply: We add references
Reviewer 2 Report
- In line 236, the treatment method in the text mainly affects the surface of large pores, and it is suggested to supplement the evidence that the morphology of smaller pores changes significantly.
- It is suggested to supplement the mechanism of pore morphology change in improving biocompatibility.
- It is recommended to revise the format of the reference to make it standardized.
Author Response
We would like to express our great appreciation to the Editor and respected Reviewers for examining the manuscript carefully and providing us valuable comments and constructive suggestions which are very important for the revision of the manuscript and improving the quality of the work. We have tried our best to revise our manuscript according to their suggestions.
Replies to Reviewer #2:
Reviewer #2:
Comment 1: In line 236, the treatment method in the text mainly affects the surface of large pores, and it is suggested to supplement the evidence that the morphology of smaller pores changes significantly.
Reply: we add recommendation to compare Fig. 6d) and 3f) on which the evidence is, in our opinion, clearly seen
Comment 2: It is suggested to supplement the mechanism of pore morphology change in improving biocompatibility.
Reply: we add comment after Your suggestion
Comment 3: It is recommended to revise the format of the reference to make it standardized.
Reply: We revised the references list
Reviewer 3 Report
In Page 2, Line 48, mention which alloy element removed in the study (say Mg instead of saying one of the elements).
Introduction may be improved by adding more literature relating thermal dealloying in Ti alloys.
MA must be expanded in the line when it appears first.
In Line 67, Add the word “purity” in Nb (99.98%), Mg (99.8%) as well, otherwise it will confuse the readers.
For implants application, bending strength of the composite is also an important mechanical property to be considered for the material design. Though authors found compression strength and elastic modulus, I suggest authors to examine and present bending strength and modulus.
Stress-strain data from compression test must be added. This will support how the mechanical properties are affected by porosity.
Moreover, to support “The moduli is close to that of human bone, so such materials have 221
a promising application potential”, please mention which human bone do you refer and what its modulus.
Results from the current results must be compared with other literature.
Conclusion has more general statements. Emphasise the research outcome.
Most of the references are very old literatures, Cite latest research within 5 years, both in Introduction and result discussion.
Some repetitions of statements may be avoided. Use of first person noun may be avoided.
Author Response
We would like to express our great appreciation to the Editor and respected Reviewers for examining the manuscript carefully and providing us valuable comments and constructive suggestions which are very important for the revision of the manuscript and improving the quality of the work. We have tried our best to revise our manuscript according to their suggestions.
Replies to Reviewer #3:
Reviewer #3:
Comment 1: In Page 2, Line 48, mention which alloy element removed in the study (say Mg instead of saying one of the elements).
Reply: improved
Comment 2: Introduction may be improved by adding more literature relating thermal dealloying in Ti alloys.
Reply: Thermal dealloying applied to Ti alloys is quite new method of foams preparation. We cite our previous study. However, we don’t want to increase the auto-citation level. Unfortunately till now we cannot find study by other authors correspond to this method.
Comment 3: MA must be expanded in the line when it appears first.
Reply: improved
Comment 4: In Line 67, Add the word “purity” in Nb (99.98%), Mg (99.8%) as well, otherwise it will confuse the readers.
Reply: improved
Comment 5: For implants application, bending strength of the composite is also an important mechanical property to be considered for the material design. Though authors found compression strength and elastic modulus, I suggest authors to examine and present bending strength and modulus.
Reply: We add more information about mechanical properties. However, after consultation, we have to inform that bending strength measurement is impossible to perform on our samples. It is very valuable suggestion and we will try to perform further mechanical investigation in the future
Comment 6: Stress-strain data from compression test must be added. This will support how the mechanical properties are affected by porosity.
Reply: We add stress-strain diagram and comment
Comment 7: Moreover, to support “The moduli is close to that of human bone, so such materials have 221 a promising application potential”, please mention which human bone do you refer and what its modulus.
Reply: improved, we add reference
Comment 8: Results from the current results must be compared with other literature.
Reply: We add comparison with other authors results and expand the discussion
Comment 9: Conclusion has more general statements. Emphasise the research outcome.
Reply: improved
Comment 10: Most of the references are very old literatures, Cite latest research within 5 years, both in Introduction and result discussion.
Reply: improved, we add more references
Comment 11: Some repetitions of statements may be avoided. Use of first person noun may be avoided.
Reply: improved where we could
Reviewer 4 Report
In the present paper, a metallic foam, based on a Titanium alloy, was produced and characterized as a possible material for medical implants. The topic is interesting for the reader and absolutely in the aims of the scope of the Journal. Only minor revisions from my side:
Introduction: this section is clear, but a more deep and detailed explanation of the state of the art must be included, as a more detailed explanation of the aims of the work and novelty with respect to previous literature must be provided (last paragraph).
Materials and Methods: please include a space between each number and its unit of measure, e.g. 1300 °C
Results and discussion: when discussing Figure 2 please indicate what are the white and black parts of the micrographs, metal or porosity? How porosity has been measured in details? Imaging elaboration? About Figure 3 please indicate from which sample these images have been taken with respect to milling time and temperature of thermal dealloying. The same for all the other figures that follows. Line 205, how theoretical porosity has been calculated? Line 207, the terms "smaller" and "bigger" are quite subjective, please indicate a reference to better understand how to identify and discriminate what is "small" from what is "big". Line 227 to 232 please include a REF or indicate from which result this statement can be drawn. In Figure 5 images of the same surface before electrochemical etching should be provided for comparison. I suggest moving Figure 6 in another position through the text, as it divides a paragraph in two parts and, thereafter, this paragraph is not easy to read. Figure 7, space lines before and after this image should be provided.
Author Response
We would like to express our great appreciation to the Editor and respected Reviewers for examining the manuscript carefully and providing us valuable comments and constructive suggestions which are very important for the revision of the manuscript and improving the quality of the work. We have tried our best to revise our manuscript according to their suggestions.
Replies to Reviewer #4:
Reviewer #4:
Comment 1: Introduction: this section is clear, but a more deep and detailed explanation of the state of the art must be included, as a more detailed explanation of the aims of the work and novelty with respect to previous literature must be provided (last paragraph).
Reply: the introduction was improved and we add more relevant references.
Comment 2: Materials and Methods: please include a space between each number and its unit of measure, e.g. 1300 °C
Reply: we improved according to your suggestions. However excluding “°C” and “%” symbols – those should be written without the space.
Comment 3: Results and discussion: when discussing Figure 2 please indicate what are the white and black parts of the micrographs, metal or porosity? How porosity has been measured in details? Imaging elaboration?
Reply: improved (in materials and methods part)
Comment 4: About Figure 3 please indicate from which sample these images have been taken with respect to milling time and temperature of thermal dealloying. The same for all the other figures that follows.
Reply: improved – we add information in text
Comment 5: Line 205, how theoretical porosity has been calculated?
Reply: to clarify: no theoretical porosity but “theoretical calculation”. We write about it in materials and methods part. However after your suggestion we change the phrase to “designed value (60%)”
Comment 6: Line 207, the terms "smaller" and "bigger" are quite subjective, please indicate a reference to better understand how to identify and discriminate what is "small" from what is "big".
Reply: The answer for Your suggestion is 4 lines above: “relatively large pores (1 to over 100 m)… smaller (approx. 0.25 to 1 m)”
Comment 7: Line 227 to 232 please include a REF or indicate from which result this statement can be drawn.
Reply: improved, we add reference
Comment 8: In Figure 5 images of the same surface before electrochemical etching should be provided for comparison.
Reply: The surface before electrochemical treatment is shown in figure 3.
Comment 9: I suggest moving Figure 6 in another position through the text, as it divides a paragraph in two parts and, thereafter, this paragraph is not easy to read.
Reply: We agree. However, after such shifting a lot of free space remains. Maybe we will wait for Editor decision
Comment 10: Figure 7, space lines before and after this image should be provided.
Reply: improved
Round 2
Reviewer 1 Report
The manuscript is revised according to the comments. Some revisions are still required. The format and language need revisions. Therefore, I recommend publication after a minor revision.